# Descriptive Epidemiology of breast and gynecological cancers among patients attending Saint Paul's Hospital Millennium Medical College, Ethiopia

**Haimanot E. Hailu**[1]*, **Alison M. Mondul**[2], **Laura S. Rozek**[2‡], **Temesgen Geleta**[1‡]

**1** Department of Public Health, Saint Paul's Hospital Millennium Medical College, AA, Ethiopia,
**2** Department of Epidemiology, School of Public Health, University of Michigan, Ann Arbor, MI, United States of America

☯ These authors contributed equally to this work.
‡ These authors also contributed equally to this work.
* metiewnetu@gmail.com

## Abstract

### Introduction

Cancer is a leading cause of death in both more and less economically developed countries; the burden is expected to grow in less developed countries, such as Ethiopia. Lack of adequate information is one of the major problems preventing the design of cancer control strategies in Ethiopia.

### Objective

To characterize gynecological and breast cancers among clients attending Gynecologic clinic of Saint Paul Hospital Millennium Medical college over 5 year period.

### Methods

We retrospectively reviewed characteristics of 2,002 female cancer patients who visited the Oncology unit of Saint Paul's Hospital Millennium Medical College from 2014–2018. We estimated the proportion, pattern and trend of common types of gynecologic cancers as well as breast cancer. The ten years incidence projection was also computed.

### Result

From the 2,002 malignancies, cervical (46.7%) was the most frequent cancer followed by breast (29.3%) and ovarian cancers (13%). The majority of breast cancers were observed among younger patients whereas cervical cancer was predominantly observed among older women. An overall increment in number of breast and gynecologic cancer was observed over the five years period.

**Data Availability Statement:** All relevant data are within the manuscript and its Supporting Information files.

**Funding:** This study is funded by Saint Paul's Hospital Millennium Medical College.

**Competing interests:** The authors have declared that no competing interests exist.

## Conclusion

In this descriptive study, we found that breast and gynecologic cancers are important public health problems among women in Addis Ababa, and that the number of patients seeking care for these cancers is increasing. Additional studies are needed to identify risk factors for these cancers, particularly among younger women, to characterize the trends over time and to project the scope of the cancer problem expected in the future to inform cancer control programs. Increasing public awareness on the possible risk factors and screening is mandatory in addition to resource allocation for further studies and targeted intervention.

## Introduction

Cancer is a prominent cause of mortality in both economically developed and underdeveloped nations. Unfortunately, the burden is expected to grow globally due to the development and aging of the population, predominantly in less developed countries, where nearly 82% of the world's population resides. Cancer has been found to increase with adoption of life style behaviors like poor dietary habit, sedentary life style, smoking, and reproductive characteristics including having low number of birth, first birth at older age, etc., have further increased the cancer burden in less economically developed countries [1]. Cancer was regarded as a problem of affluent societies in the past, however the recent report from GLOBOCAN incidence estimates indicates that greater than 50% of all new cancer cases occur in third world countries (2,561,666 cases compared with 2,175,974 cases in the more developed world)[2].

Gynecological cancer, which includes cancers of the cervix, ovary, uterus, vulva, vagina and fallopian tube are among the leading causes of cancer-related mortality worldwide and the distribution and frequency vary across regions. These cancers account for about 10% of all cancers diagnosed in women. The proportion of female genital cancers range from 31.6–35.0% in sub-Saharan Africa to 12.7–13.4% in North America [2, 3]. In 2000, 57,000 of gynecologic cancer cases were reported from sub-Saharan Africa which accounts for 22% of all female cancers corresponding to an age-adjusted incidence of 31/100,000. The disease is among the major causes of cancer related mortality in the continent which is mainly due to late presentation and poor access to diagnosis and treatment [4, 5]. In the year 2010, Ethiopia had a projected population of 20.9 million women aged 15 years and older who were at risk of developing cervical cancer. In the same year, the age-specific incidence of cervical cancer in Ethiopia was higher than the world average for women ages 55 and above. According to a recent WHO report, cervical cancer ranks as the second most common cancer among women in Ethiopia [6].

Although gynecologic cancers are an important public health problem, as discussed above, breast cancer is the most common malignancy in women accounting for more than $1/5^{th}$ of the global burden of cancers worldwide. In 1990s, a 33% increment of breast cancer cases was observed which accounts for more than a million new cases in recorded in 2000. In less developed world, it is the first leading cause of mortality among women accounting for 324,000 deaths that represents more than 14% of all deaths [7]. In Ethiopia breast cancer was the most common malignancy accounting for $1/3^{rd}$ of the cancers in women [8].

Information about cancer related deaths and estimation of new/existing number of cancer cases are the most important epidemiological actions that are mandatory to design appropriate cancer control activities and inform the design of national health programs. Cancer incidence

estimation was conducted in Ethiopia using population based cancer registry data characterizing recent trends (i.e. 2012–2015) for the most common types of cancer [8]. Our study will supplement this existing information by indicating how the incidence of cancer is changing over a longer and more updated time period (i.e. 2014–2018), and providing more detailed information including histological characterization of tumors. Once the varied epidemiological profiles of female cancers in our population are characterized, this information can guide policy makers in determining necessary cancer control measures. To fill the existing information gap, we have used Saint Paul's Hospital Millennium Medical College (SPHMMC)'s oncology unit data to characterize breast and gynecologic malignancies.

## Methods

We conducted a retrospective review of Oncology unit records of patients diagnosed to have either breast or any of the gynecological malignancies between January 2014 and December 2018.

SPHMMC is located in Addis Ababa, the capital city of Ethiopia. It is a teaching and referral Hospital located in the western part of Addis Ababa. It is the second hospital offering cancer treatment in the country. All patients who were seen for suspicion of cancer or who were diagnosed with cancer at the Oncology unit and who registered with full information including age, marital status, place of residence, diagnosis in the registration book or in the chart were considered to be eligible for the study.

The following information was collected from the medical record: age, marital status, residence, stage at diagnosis, and cancer histopathological type. Histologic diagnosis was conducted by the Oncology unit of the hospital. The data was analyzed using SPSS version 25.0 (SPSS Inc., Chicago, IL, USA). Frequency distributions of socio-demographic and tumor characteristics of study participants were generated. Linear regression was conducted to identify whether the increase in number of new cases over years is statistically significant and $P<0.05$ was considered the threshold for statistical significance.

This study was approved by the SPHMMC Institutional review Board. An official letter of permission was obtained from the Hospital to access the data from the record of patients. The obtained information was kept confidential and only be used for research purpose.

## Result

The total number of female patients who visited the Oncology unit of SPHMMC over the five year period from 2014–2018 with the suspicion of cancer was 9,261. Five thousand three hundred forty five of these patients were found to have either a malignant or benign tumor, with 2,002 (37.5%) of them being malignant.

Of the total number of cancer cases (n = 2,002), the majority of patients were within the age range of 35–39 and 45–49 accounting for 311 (15.5%) and 265 (13.2%) cases, respectively. The mean and median ages of all the patients were 47.05 ± 13.5 and 46, respectively. The vast majority of the patients were married (>90%).

### Demographic patterns

We found that cervical cancer was the most common type of cancer diagnosed [n = 935 (46.7%)] followed by breast [n = 587(29.3%)] and ovarian cancers [n = 260(13%)] (Fig 1). Breast cancer was most common among patients aged 35–39 and dramatically declined after the age of 50 years (Fig 2). In contrast, the risk of cervical cancer appeared to generally increase with age, with the risk plateauing around age 50–54 (Fig 2). Additionally, the number of cases of other types of malignancies (i.e. ovarian, endometrial and myometrial cancers) appeared

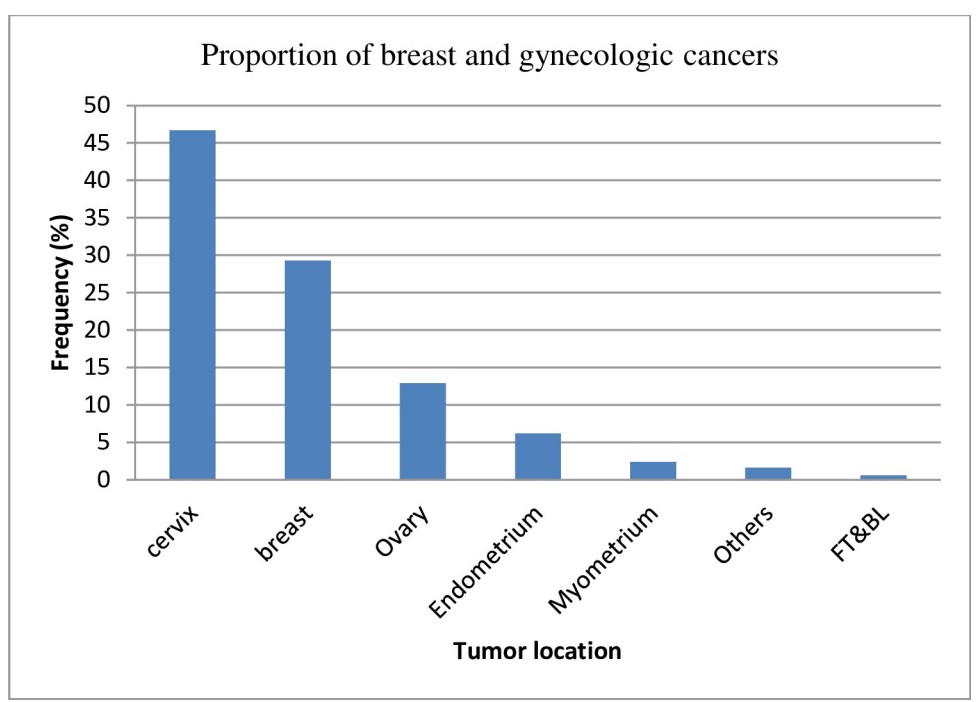

**Fig 1. Proportional morbidity of breast and gynecological cancers among clients presenting at SPHMMC, 2014–18.** FT&LB: Fallopian tube and Broad ligaments.

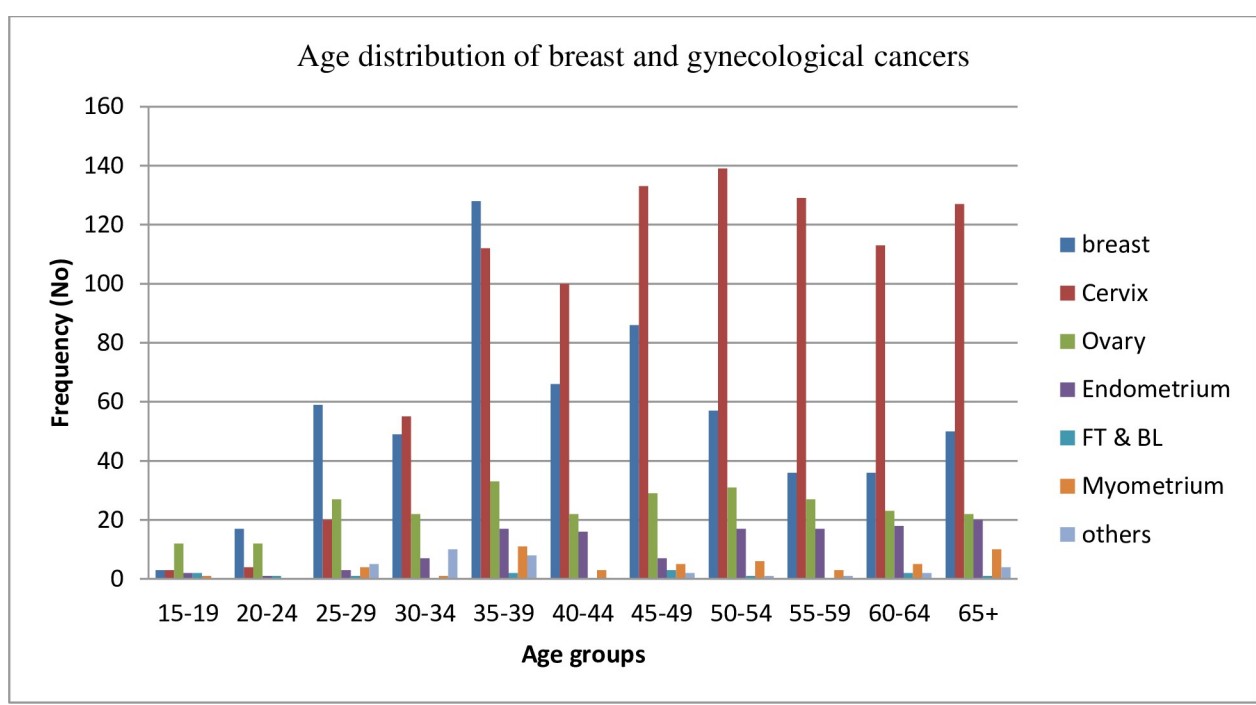

**Fig 2. Age distribution of breast and gynecological cancers presenting at SPHMMC, 2014–2018.**

**Table 1. Distribution of breast and gynecological cancers presenting at SPHMMC by place of residence 2014–18.**

| Place of residence | Frequency No (%) |
|---|---|
| Addis Ababa | 1098(54.8%) |
| Oromia | 763(38.1%) |
| Amhara | 126(6.3%) |
| Tigrai | 15(0.75) |
| Total | 2002(100%) |

not to increase markedly with age (Fig 2). More than half of the patients reported that they were from Addis Ababa: breast n = 332 (56.6%), cervix n = 506 (54.1%), and ovary n = 136 (52.3%). The Oromia and Amhara regions were the next two common regions represented, reflecting their increasing distances from the Saint Paul's Hospital Millennium Medical College (Table 1).

## Histopathologic subtypes

There were a total of 587 histologically confirmed breast cancer cases with different histologic types broadly classified as carcinoma [n = 570 (97.1%)], sarcoma [n = 14 (2.4%)] and phyllodes tumor [n = 3 (0.5%)]. Among the carcinoma cases, ductal carcinoma (81.6%) was the predominant subtype of cancer observed (Table 2). Squamous cell carcinoma of the cervix was the most common type of cervical malignancy detected accounting for 90.% of all cases, while adenocarcinoma (3.85%) was the second most observed type of cervical cancer (Table 3). For ovarian cancer cases, serous tumors were the most common histopathologic type, accounting for 37.3% of ovarian cancers. Mucinous tumors were the next most common histopathologic type accounting for 11.1% of cases (Table 4). Among the cancers of endometrium, squamous cell carcinoma was the most common type constituting more than a quarter (27.2%) of the cases followed by adenocarcinoma which accounted for 18.4%. Stromal carcinoma and leiomyosarcoma were among the types with few cases (Fig 3).

## Time trends

The number of breast and gynecologic cancers being treated at SPHMMC increased over the five year period from 2014–18. A slight decrement was observed in the year 2017 where 390 new cases were diagnosed. However, the number of female breast and gynecologic cases increased the following year with 554 new cases diagnosed in 2018 (Fig 4).

**Table 2. Histopathological classification of breast cancer, 2014–18, SPHMMC.**

| Histopathological type | Frequency (%) |
|---|---|
| Carcinoma | 570(97.1%) |
| Adenocarcinoma | 7(1.2%) |
| Unspecified carcinoma | 40(6.81%) |
| Ductal | 479(81.6%) |
| Lobular | 23(3.9%) |
| Mixed | 17(2.9%) |
| Metaplastic | 4(0.7%) |
| Sarcoma | 14(2.4%) |
| Phyllodes tumor | 3(0.5%) |
| Total | 587(100%) |

Table 3. Histopathological classification of cervical cancer, 2014–18, SPHMMC.

| Histopathogic type | Frequency (%) |
| --- | --- |
| Squamous cell carcinoma | 844(90.3%) |
| Adenocarcinoma | 36(3.85%) |
| Adenosquamous carcinoma | 11(1.2%) |
| Carcinoma, unspecified | 22(2.35%) |
| Others | 22(2.35%) |
| Total | 935(100%) |

Based on this trajectory, the number of breast and gynecologic cancers seen at SPHMMC are estimated to continue increasing over the next decade as estimated based on method of best fit (least squares) linear regression. By 2028, the numbers of new cases are expected to double from 2018, increasing from 554 new cases to 1141 new cases per year ($R^2 = 0.98$).

## Discussion

Globally, gynecological malignancies are among the top causes of cancer related mortality even though the distribution and frequency is variable across regions [2, 3]. Our study demonstrates that out of total gynecological malignancies, cervical cancer was the most common with breast cancer the second most common cancer observed, accounting for more than 75% of the total malignancies observed over a five year period. These findings are similar to one previous study conducted in Tikur Anbessa Specialized Hospital, Addis Ababa [9], which indicated that gynecological and breast cancer were the two most common types of malignancies observed accounting for 58.86% of total cancer cases. Our findings are also similar to those from a study conducted in Pakistan [10] and a recent IARC report indicating that cervical cancer is the commonest gynecological malignancy in developing countries where organized screening programs do not exist [11]. The high number of cervical cancer cases in Ethiopia can be explained by a high rate of early marriage and a high number of live births [12] which might indicate early sexual commencement and possible change in partner, absence of a routine cervical

Table 4. Histopathological classification of ovarian cancer, 2014–18, SPHMMC.

| Histopathologic type | Frequency (%) |
| --- | --- |
| Serous (n = 97) | 97(37.35) |
| Carcinoma | 45(46.4%) |
| Adenocarcinoma | 27(27.83) |
| Cystadenocarcinoma | 12(12.37) |
| Papillary Carcinoma | 13(13.4%) |
| Mucinous(n = 29) | 29(11.15) |
| Carcinoma | 10(34.5%) |
| Adenocarcinoma | 9(31%) |
| Cystadenocarcinoma | 10(34.5%) |
| Teratoma | 22(8.46%) |
| Squamous cell carcinoma | 17 (6.5%) |
| Carcinoma, unspecified | 17(6.5%) |
| Adenocarcinoma | 13(5%) |
| Cystadenocarcinoma | 9(3.5%) |
| Others | 56 (21.6%) |
| Total | 260 |

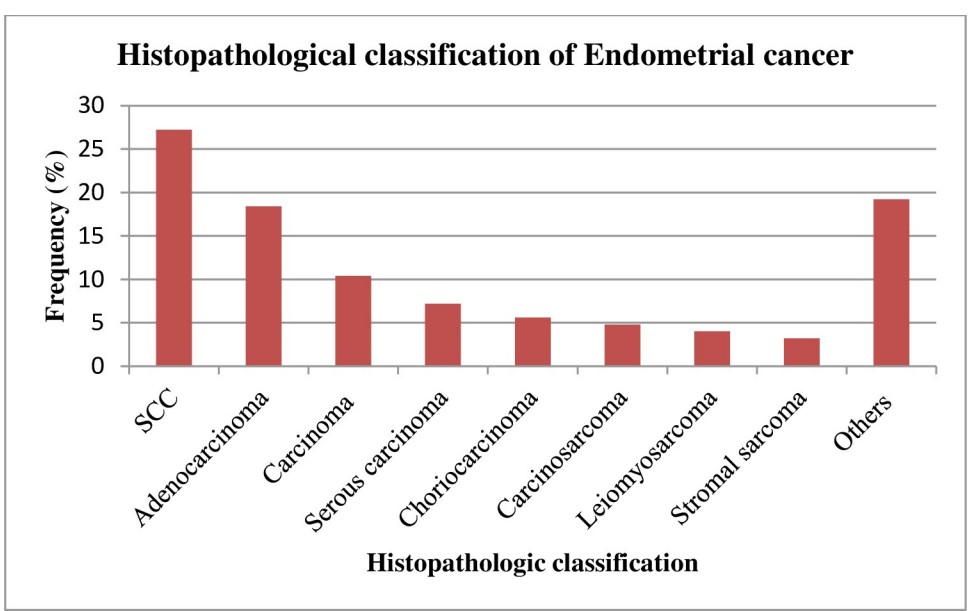

**Fig 3. Histopathological classification of endometrial cancer, 2014–18, SPHMMC.** SCC: Squamous cell carcinoma.

cancer screening program, and low uptake of screening among high risk populations [13]. However, this finding is contrary to the estimates of cancer incidence in Ethiopia in 2015 using population based registry data [8] where breast cancer was found to be the commonest cancer followed by cervical and ovarian malignancies. This difference might be explained by the difference in sample size or by differences in prevalence by region as the former study used

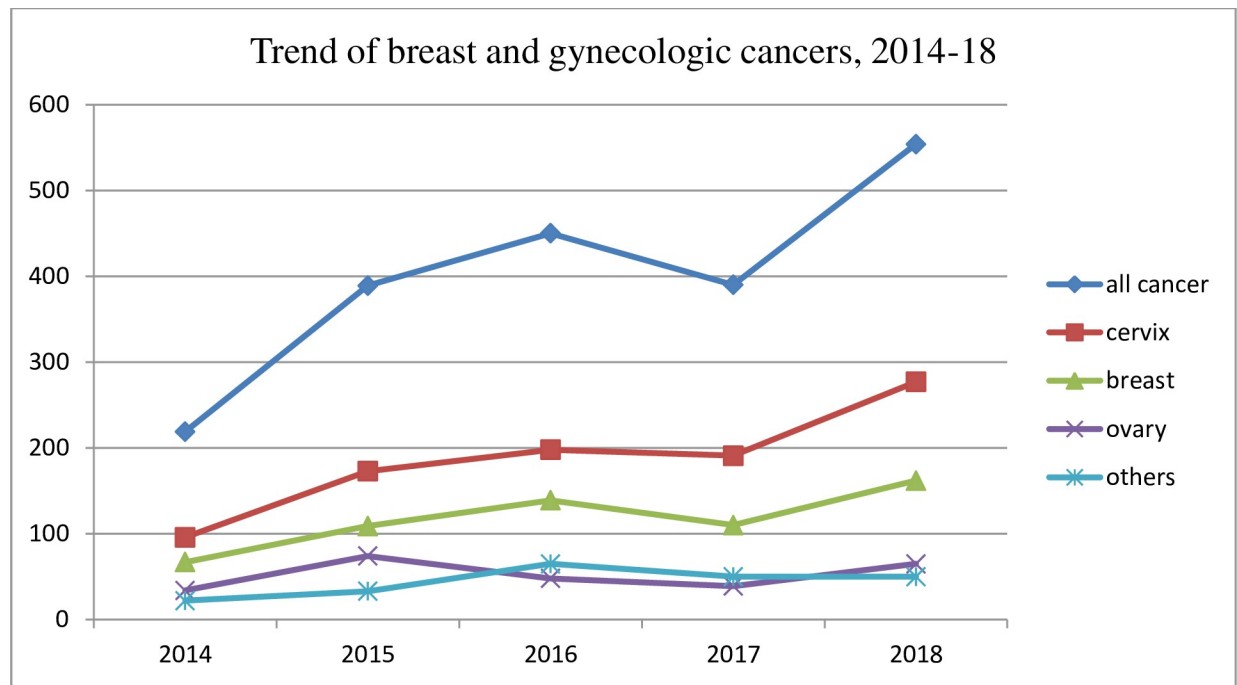

**Fig 4. Trend of breast and gynecologic cancers, 2014–18, SPHMMC.**

hospital based data collected from six regions in the country in addition to Addis Ababa cancer registry data.

Most of the tumors in our study (i.e. breast, ovarian and endometrial cancers) are most frequently diagnosed among younger women, which is similar to most studies conducted in Sub Saharan Africa [7, 9] but in contrast to findings from western population based studies [14, 15] [16]. This could possibly be explained by better health seeking behavior among younger women in Ethiopia where older women may not seek care for cancer. Another possible explanation could be the presence of environmental risk factors for these cancers that may have increased in recent time periods. In addition, genetic risk factors and/or the interplay of genetic and environmental or lifestyle factors may play a role. Brinton et al. [17] also elaborated the fact that African women may develop unique breast tumor subtypes could also be an important contributory factor to the unusual age distribution noted in Africa. Further work is needed to understand the apparent higher risk of these cancers among younger women in Ethiopia.

The fact that the incidence of cervical cancer increases in older age groups in our study is similar to others conducted in Addis Ababa [18] and Nigeria [19]. This might be due to late presentation of patients at an advanced clinical stages of the disease or not receiving or completing prescribed courses of treatment due to deficiencies in treatment availability, accessibility, and affordability as demonstrated in other similar studies [20]. Our data suggest that women in our study develop cervical cancer at later ages compared to European populations, where the incidence of cervical cancer begins to increase at age 20–29 years and upsurges rapidly to reach a highest level usually around age 45–49 years [21]. However, data from the United States Surveillance, Epidemiology and End Results (SEER) registry system in 2010 demonstrated that, even in the US, blacks are older than whites at diagnosis for cervical cancer among others which is explained to be determined by etiologic, subtype heterogeneity as well as disparities in access to care [22].

Our findings by histopathologic subtype were very similar to those previously reported in the literature for multiple populations worldwide. For breast cancer, we observed that ductal carcinoma was the most commonly observed histopathologic type, which is similar to the findings from other studies conducted among African women [23–25] and in the United states as well [26]. It should be noted, however, that this might be influenced by different factors including the difficulty of evaluating the prevalence of more unusual types of breast cancer, and the difficulty of determining the histopathology of advanced-stage, poorly differentiated tumors that have lost any distinguishing features, which are more common in African populations.

We have found squamous cell carcinoma of the cervix to be the most commonly observed histolopathologic type which is similar to other studies conducted in Pakistan and Bangladesh [10] [27]. The fact that this histopathologic type is pathogenically associated with Human Papilloma Virus might explain its prevalence. However, some studies indicate that adenocarcinoma, increasing and up to 20% to 25% is reported in some cases [28]. The same finding is observed in our study where its incidence increased from 8% in 2014 to 35% in 2018.

Epithelial, specifically serous tumor, was the predominant type of ovarian cancer in our population, similar to the findings of studies conducted in other populations [10, 27, 29]. Unfortunately, serous tumors are mostly high-grade serous carcinomas and are characterized by aggressive behavior, late-stage diagnosis, and low survival, contributing to the poor survival for ovarian cancer overall [30].

The overall incidence of breast and gynecologic cancer increased over the five year study period with a minimal downturn in the year 2017; this might have been due to the overall decrease in the number of hospital visitors due to social instability that occurred in Ethiopia during 2017. The observed increase in incidence is probably due to changes in socio

demography, lifestyle or increased awareness of patients about cancer and care seeking behavior. The increasing trends in the number of breast and gynecologic cancer patients in our study area is similar to those observed across Africa [9, 18], but differs from trends in developed countries where the incidence and mortality due to cancer is decreasing[31, 32]. This difference is largely due to differences in exposures to major risk factors, detection practices like availability of diagnostic and screening services, awareness of early signs and symptoms, and availability of treatment.

To our knowledge, this study is the first to characterize female breast and gynecologic cancers in the country with respect to providing more recent trend and histopathologic information. However, our study had a few limitations. We lacked information on some important variables, including stage of the malignancy, as well as other demographic and lifestyle variables as these were not recorded in the medical records. Additionally, having data from a limited number of years makes it difficult to run most robust analysis methods for trends across time, such as joint point analysis.

## Conclusion

In conclusion, in this descriptive study we found that breast and gynecologic cancers are important public health problems among women in Addis Ababa, and that the number of patients seeking care for these cancers is increasing. Importantly, many of these cancers were more common in younger women than expected, based on studies conducted in Western populations.

General, nonspecific prevention mechanisms including change in life styles and early screening would help to minimize risk of majority of breast and gynecological cancers. However, specific measures like mastectomy may prevent breast cancer while cervical cancer can be prevented with safe sexual practices and vaccination. Therefore, there is the need to increase public awareness and consecutive practices of these factors. Government and non-governmental organizations should establish sustainable national and regional cancer screening programs and treatment centers.

Additional studies should be conducted to identify the possible risk factors for these cancers, particularly among younger women, to characterize the trends over time and to project the scope of the cancer problem expected in the future. This information will be invaluable in informing the creation of a comprehensive cancer control plan and targeted interventions for the prevention of breast and gynecologic cancers in Ethiopia.

## Supporting information

**S1 Dataset.**
(XLSX)

## Acknowledgments

We thank the Saint Paul's Hospital Hospital Millennium Medical College oncology unit and patient registry department for their cooperation.

## Author Contributions

**Data curation:** Haimanot E. Hailu, Temesgen Geleta.

**Formal analysis:** Haimanot E. Hailu, Alison M. Mondul, Laura S. Rozek, Temesgen Geleta.

**Funding acquisition:** Haimanot E. Hailu.

**Investigation:** Haimanot E. Hailu, Temesgen Geleta.

**Methodology:** Haimanot E. Hailu, Alison M. Mondul, Laura S. Rozek, Temesgen Geleta.

**Project administration:** Haimanot E. Hailu.

**Software:** Haimanot E. Hailu, Alison M. Mondul, Laura S. Rozek.

**Supervision:** Alison M. Mondul, Laura S. Rozek.

**Writing – original draft:** Haimanot E. Hailu, Alison M. Mondul, Temesgen Geleta.

**Writing – review & editing:** Haimanot E. Hailu, Alison M. Mondul, Laura S. Rozek, Temesgen Geleta.

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
