## [Decision Letter · Decision Letter 0]

11 Dec 2019

PONE-D-19-27493

Descriptive Epidemiology of breast and gynecological cancer among patients attending Saint Paul’s Hospital Millennium Medical College, Ethiopia

PLOS ONE

Dear Dr Haimanot Hailu,

Thank you for submitting your manuscript to PLOS ONE. After careful consideration, we feel that it has merit but does not fully meet PLOS ONE’s publication criteria as it currently stands. Therefore, we invite you to submit a revised version of the manuscript that addresses the points raised during the review process.

Please consider carefully the detailed comments of the reviewers and revise the presentation of the statistical analyses according to the comments and other suggestions of Reviewer 1. Please also consider the critical comments of Reviewer 2 concerning the patient material and  the prevention, mechanisms and clinical characteristics of breast and gynecological cancers and revise the manuscript accordingly  that the acceptance of the manuscript could be reconsidered.

We would appreciate receiving your revised manuscript by January 31, 2020. To enhance the reproducibility of your results, we recommend that if applicable you deposit your laboratory protocols in protocols.io, where a protocol can be assigned its own identifier (DOI) such that it can be cited independently in the future. For instructions see: http://journals.plos.org/plosone/s/submission-guidelines#loc-laboratory-protocols

We look forward to receiving your revised manuscript.

Kind regards,

Pirkko L. Härkönen, M.D., Ph.D.

Academic Editor

PLOS ONE

Journal Requirements:

1)

2)

We suggest you thoroughly copyedit your manuscript for language usage, spelling, and grammar. If you do not know anyone who can help you do this, you may wish to consider employing a professional scientific editing service.  

3)

We noticed minor instances of text overlap with the following previous publication(s), which need to be addressed:

https://onlinelibrary.wiley.com/doi/full/10.3322/caac.21262

https://obgyn.onlinelibrary.wiley.com/doi/full/10.1016/S0020-7292%2803%2900225-X

http://etd.aau.edu.et/bitstream/handle/123456789/8425/1.TEFERI%20ATAKILTI.pdf?sequence=1

The text that needs to be addressed involves the Introduction section.

In your revision please ensure you cite all your sources (including your own works), and quote or rephrase any duplicated text outside the methods section. Further consideration is dependent on these concerns being addressed.

4)

In the ethics statement in the manuscript and in the online submission form, please provide additional information about the patient records used in your retrospective study. Specifically, please ensure that you have discussed whether all data were fully anonymized before you accessed them and/or whether the IRB or ethics committee waived the requirement for informed consent. If patients provided informed written consent to have data from their medical records used in research, please include this information.

Reviewers' comments:

Reviewer's Responses to Questions

**Comments to the Author**

1. Is the manuscript technically sound, and do the data support the conclusions?

Reviewer #1: Yes

Reviewer #2: No

2. Has the statistical analysis been performed appropriately and rigorously? 

Reviewer #1: Yes

Reviewer #2: Yes

3. Have the authors made all data underlying the findings in their manuscript fully available?

Reviewer #1: Yes

Reviewer #2: Yes

4. Is the manuscript presented in an intelligible fashion and written in standard English?

Reviewer #1: Yes

Reviewer #2: No

5. Review Comments to the Author

Reviewer #1: The manuscript has been review and found to have merit for publication. The authors should address the issues identified and re-submit:

Gynecological cancer in the title should read: "gynecological cancers" instead of cancer.

In figure 1, cervical cancer which has the highest frequency was omitted. Kindly add that of cervical cancer cases. Secondly the title for figure 1 is not appropriate as these measures presented are not prevalence but rather proportional morbidity of breast and gynecological cancers presenting at SPHMMC. Change the title of the figure and write the appropriate title for figure 1.

Figure 2, the line graph used for presentation is not appropriate. The author should used a multi-bar graph to show the age distribution since this is not a time trend but rather age distribution. The title should change to the age distribution of breast and gynecological cancers presenting at SPHMMC, 2014 - 2018

There is no table or figure to show the distribution of the cases by place. I suggest the authors show the distribution by place using a spot map or any other appropriate presentation.

The authors indicated that, by 2028, the numbers of new cases are expected to double from 2018, increasing from 554 new cases to 1141 new cases per year (Figure 4). However the calculation or model used to arrive at that is not shown in the methods or results. The authors should clearly show the methodology and assumption in arriving at that projected figure.

The authors also indicated in the last paragraph of the results that,the association between the number of new cases and time was marginally statistically significant (P=0.054). I suggest this statement should be deleted as the p-value of 0.054 is not significant and that statement add nothing relevant to the results.

Reviewer #2: 1. It helps the reviewer, if page and line numbers would be placed

2. Semoraphic patterns: As to Oroma and Amtiara regions, do the figures of lower numbers of cancer cases reflect the distance to the hospital or rather a lower population.

3. Time trends: does the increase of cancer cases mean a real increase of cases or possibly better diagnotics and inceased remittance.

4. Discussion: I do not understand, how the early marriage or bigger number of life birts explain the high nb of cases of cervical cancer. Cervical cancer is a sexually transmitted infection (HP-virus) which is almost always involved in the development of cervical cancer. The possibility for transmission depends on nb of ontercourses without condom, nb of partners etc.

5. I do not understand what environmental factors you mean.

Last page: "Breast cancer and gynecological cancers are largely preventable diseases..." Cases of breast cancer can largely be prevented with mastectomies (in those 5-10% of women who have hereditary BRCA1 or 2 genes). Cervical cancer can be prevented by rigorous use of condoms or a vaccination before initiation of coital activities. The other gynecological cancers can not be prevented, except by surgically removing the gyn organs. Instead, with good early diagnostics and adequate forms of therapies, the number of deaths due to cancers can be prevented.

6. PLOS authors have the option to publish the peer review history of their article (what does this mean?). If published, this will include your full peer review and any attached files.

Reviewer #1: Yes: Kofi Nyarko

Reviewer #2: No

---

## [Author Response · Author response to Decision Letter 0]

26 Jan 2020

We thank the reviewers for their valuable comments. The point by point response to reviewers is attached.

---

## [Editor Report · Decision Letter 1]

26 Feb 2020

PONE-D-19-27493R1

Descriptive Epidemiology of breast and gynecological cancers among patients attending Saint Paul’s Hospital Millennium Medical College, Ethiopia

PLOS ONE

Dear Dr Haimanot Hailu,

Thank you for submitting your revised manuscript to PLOS ONE. The revised version addresses most of the points raised during the review process in an acceptable way. However, I would like to point out one concern and question  (point 5) raised by Reviewer 2.

The comment concerned one sentence in the "Conclusions" part: "Breast and gynecological cancers are largely preventable diseases".  The reviewer wanted to get this sentence justified and offered explanations. Although your reply is generally acceptable, it did not make the sentence more clear. Although breast cancer is generally considered to be largely influenced by the environmental factors, diet, life style, reproductive history etc,, there are no specific means to prevent the disease at the individual level. The same is true in case of ovarian cancer and other gynecological cancers. On the contrary, cervical cancer, besides the mode of sexual behaviour, can be be prevented to a great extent by vaccination. Therefore I would ask you to reconsider this sentence and to specify how and to which extent breast cancer and other gynecological cancers can be prevented.

We would appreciate receiving your revised manuscript by March 9, 2020. To enhance the reproducibility of your results, we recommend that if applicable you deposit your laboratory protocols in protocols.io, where a protocol can be assigned its own identifier (DOI) such that it can be cited independently in the future. For instructions see: http://journals.plos.org/plosone/s/submission-guidelines#loc-laboratory-protocols

We look forward to receiving your revised manuscript.

Kind regards,

Pirkko L. Härkönen, M.D., Ph.D.

Academic Editor

PLOS ONE

---

## [Author Response · Author response to Decision Letter 1]

4 Mar 2020

Response: Thank you for your comment, its well understood. We have reconsidered and restated the sentence as follows. “General, nonspecific prevention mechanisms including change in life styles and early screening would help to minimize risk of majority of breast and gynecological cancers. However, specific measures like mastectomy may prevent breast cancer while cervical cancer can be prevented with safe sexual practices and vaccination.”

---

## [Editor Report · Decision Letter 2]

5 Mar 2020

Descriptive Epidemiology of breast and gynecological cancers among patients attending Saint Paul’s Hospital Millennium Medical College, Ethiopia

PONE-D-19-27493R2

Dear Dr. Hailu,

We are pleased to inform you that your manuscript revised as reported has been judged scientifically suitable for publication and will be formally accepted for publication once it complies with all outstanding technical requirements.

With kind regards,

Pirkko L. Härkönen, M.D., Ph.D.

Academic Editor

PLOS ONE
---

## [Editor Report · Acceptance letter]

9 Mar 2020

PONE-D-19-27493R2 

Descriptive Epidemiology of breast and gynecological cancers among patients attending Saint Paul’s Hospital Millennium Medical College, Ethiopia 

Dear Dr. Hailu:

I am pleased to inform you that your manuscript has been deemed suitable for publication in PLOS ONE. Congratulations! Your manuscript is now with our production department. 

With kind regards,

on behalf of

Dr. Pirkko L. Härkönen 

Academic Editor

PLOS ONE